# Beliefs of UK Transplant Recipients about Living Kidney Donation and Transplantation: Findings from a Multicentre Questionnaire-Based Case–Control Study

**DOI:** 10.3390/jcm9010031

**Published:** 2019-12-21

**Authors:** Pippa K. Bailey, Fergus J. Caskey, Stephanie MacNeill, Charles Tomson, Frank J. M. F. Dor, Yoav Ben-Shlomo

**Affiliations:** 1Population Health Sciences, Bristol Medical School, University of Bristol, Bristol BS8 2PS, UK; fergus.caskey@bristol.ac.uk (F.J.C.); stephanie.macneill@bristol.ac.uk (S.M.); y.ben-shlomo@bristol.ac.uk (Y.B.-S.); 2Southmead Hospital, North Bristol NHS Trust, Bristol BS10 5NB, UK; 3The Newcastle upon Tyne Hospitals NHS Foundation Trust, Newcastle upon Tyne NE7 7DN, UK; ctomson@doctors.org.uk; 4Imperial College Healthcare NHS Trust, London W12 0HS, UK; frank.dor@nhs.net

**Keywords:** living kidney donation, living-donor kidney transplantation, beliefs, inequity

## Abstract

Differing beliefs about the acceptability of living-donor kidney transplants (LDKTs) have been proposed as explaining age, ethnic and socioeconomic disparities in their uptake. We investigated whether certain patient groups hold beliefs incompatible with LDKTs. This questionnaire-based case–control study was based at 14 hospitals in the United Kingdom. Participants were adults transplanted between 1 April 2013 and 31 March 2017. LDKT recipients were compared to deceased-donor kidney transplant (DDKT) recipients. Beliefs were determined by the direction and strength of agreement with ten statements. Multivariable logistic regression was used to investigate the association between beliefs and LDKT versus DDKT. Sex, age, ethnicity, religion, and education were investigated as predictors of beliefs. A total of 1240 questionnaires were returned (40% response). DDKT and LDKT recipients responded in the same direction for 9/10 statements. A greater strength of agreement with statements concerning the ‘positive psychosocial effects’ of living kidney donation predicted having an LDKT over a DDKT. Older age, Black, Asian and Minority Ethnic (BAME) group ethnicity, and having a religion other than Christianity were associated with greater degree of uncertainty regarding a number of statements, but there was no evidence that individuals in these groups hold strong beliefs against living kidney donation and transplantation. Interventions should address uncertainty, to increase LDKT activity in these groups.

## 1. Introduction

Living-donor kidney transplantation offers the best treatment in terms of life-expectancy and quality of life [1,2,3,4,5,6] for most people with kidney failure. The healthcare costs associated with living-donor kidney transplants (LDKTs) are less than for dialysis or deceased-donor kidney transplants (DDKTs) [7,8]. The medium-term risks of donating a kidney are small [9,10,11,12] and the quality of life of donors returns to pre-donation levels after donation [13,14].

Only 20% of those listed on the UK national transplant waiting list receive an LDKT each year [15]. Certain individuals with renal disease appear to be disadvantaged: people from Black and Asian ethnic groups in the UK are less likely to receive an LDKT when compared to White people with kidney disease [16,17]. Socioeconomic deprivation is also associated with reduced access to living-donor kidney transplantation [16,17]. Older people with kidney disease are less likely to receive an LDKT when compared to younger patients [17], and women are less likely to receive an LDKT when compared to men [18,19]. Ensuring equity in living-donor kidney transplantation has been highlighted as a UK and international research priority by patients and clinicians [20,21,22]. Differing beliefs in the acceptability of living kidney donation and transplantation have been proposed as a possible explanation for the observed differences in access [17,23,24].

In this questionnaire-based case–control study, we compared the beliefs of LDKT and DDKT recipients about the acceptability of living kidney donation and transplantation. We investigated whether beliefs about living-donor kidney transplantation were associated with an individual’s sex, age, ethnicity, religion and education. We aimed to identify groups with beliefs against living-donor kidney transplantation, that may explain the observed disparities in the uptake of LDKTs.

## 2. Experimental Section

### 2.1. Participants

The study was based at 14 UK hospitals (listed in Appendix A). We wanted to investigate beliefs about living-donor kidney transplantation specifically, and not kidney transplantation in general. Therefore, we did not invite people with Chronic Kidney Disease or those on dialysis to participate, as some of these individuals may have held beliefs against transplantation in general, as opposed to living-donor kidney transplantation specifically. We obtained from each site an anonymised list of all individuals who received kidney transplants between 1 April 2013 and 31 March 2017, stratified by LDKT and DDKT status. Individuals aged <18 years at the time of transplantation, and individuals lacking mental capacity according to the Mental Capacity Act 2005, were excluded. We performed stratified random sampling using Stata 15 [25] to select, on average, 110 LDKTs and 110 DDKTs from each site, weighted by the number of transplants performed annually at each study site. Sex and 5-year age group strata-matched sampling was used to ensure a similar sample distribution by age and sex. The case–control study was designed to detect a 7-point difference in a continuous measure of patient activation (analysis of this variable not presented here) between LDKT cases and DDKT controls with 90% power, assuming a 5% significance level. The calculation indicated that 170 patients would be needed, and that, therefore, a total of 944 would be needed to allow analyses stratified by Index of Multiple Deprivation rank quintile and allow for 10% missing data. This sample size allows for the detection of a far smaller difference (0.16 Standard Deviation) for a dichotomous exposure or between 6–8% for a categorical outcome [26].

### 2.2. Questionnaire Content and Survey Tools

Paper questionnaires were mailed by post to participants by research collaborators at the study sites. Questionnaires were accompanied by a patient information sheet, an invitation letter and a return postage paid envelope. A website-address was provided so that participants could complete the questionnaire online if preferred. Non-responders were sent a second questionnaire after 4–6 weeks. Anonymised data were extracted from returned paper questionnaires at the University of Bristol and uploaded onto a secure REDCap database [27].

Transplant beliefs were assessed using questions developed by Stothers et al. [28,29]. In development, the questions were reviewed by three expert focus groups, then evaluated in a pilot study to test content reliability and validity [28]. Test–retest analysis was reported as demonstrating excellent internal consistency, and there was no evidence of ‘skew’ or ‘halo’ effects (an overall perception/feeling of satisfaction that influences all responses rather than allowing a thoughtful consideration of each individual question) [28]. Participants were asked to read ten statements describing a belief regarding living-donor kidney transplantation (Box 1). These included statements regarding the acceptability of receiving a donated kidney, coercion or pressure on family to donate, rewards for the donor, required closeness of relationship, the subsequent effect on relationship, beliefs about recipients asking family to donate, donation from offspring to parents, and the risks of donation. Participants were asked to tick one of the following options: (i) Strongly disagree, (ii) Disagree, (iii) Agree, (iv) Strongly agree, (v) Don’t know.

Box 1Belief statements.
It is morally acceptable to take a kidney from a healthy person.Donors often agree to donate due to feelings of guilt or family pressure.Donating a kidney is a rewarding experience for the live donors.Donating a kidney to someone requires an extremely close personal relationship.A living-donor kidney transplant may strengthen the relationship between the donor and recipient.Approaching a potential donor who then says no will change the relationship between the two people.Asking someone to donate makes the recipient seem selfish.It is acceptable for a parent to receive a kidney from his/her child (over 18 years old).Decisions about donation should be made by the donor alone. The recipient should not ask for a kidney.Since the donor operation is not risk free, someone who needs a kidney transplant should wait for a kidney from someone who has died.


Questionnaires assessed participant demographics as indicated in Box 2.

Box 2Participant demographic data collected.
Sex○Male; Female10-year age group○10–19 years; 20–29 years; 30–39 years; 40–49 years; 50–59 years; 60–69 years; 70–79 years; 80–89 yearsReligion○No religion; Christian; Muslim; Jewish; Hindu; Sikh; BuddhismSocioeconomic position○No formal education; Primary school; Secondary school; Vocational/Technical; University—undergraduate; University—postgraduate; OtherEthnicity coded using the UK’s Office for National Statistics 2011 census categories [30]○White; ○Asian/Asian British;○Black/African/Caribbean/Black British;○Mixed/Multiple (White and Black Caribbean, White and Black African, Any other Mixed/Multiple ethnic background);○Other (Arab, Any other ethnic group)


### 2.3. Statistical Analysis

We compared demographic characteristics between DDKT and LDKT recipients using chi^2^ tests. The proportion of DDKT and LDKT recipients selecting each level of agreement with a belief statement was calculated and initially compared using chi^2^ tests. We used multivariable logistic regression to look at the association of transplant type (LDKT versus DDKT) with a recipient’s agreement with a belief statement. For the multivariable logistic regression, the response options were coded 1–4 (1 = strongly disagree, 2 = disagree, 3 = agree, 4 = strongly agree) with ‘Don’t know’ coded as missing. For each belief statement we ran an unadjusted model and one adjusted for potential confounders. We specified, a priori, potential confounders including sex, age, education level, ethnicity and religion. We used robust standard errors to account for clustering within renal centres. Statistical analyses were performed in Stata 15 [25].

Basic descriptive statistical tests (chi^2^ tests) then were performed to look for differences in response (agreement = strongly agreed and agreed; disagreement = strongly disagreed and disagreed; and don’t know) across different patient demographic groups. For these analyses, age was dichotomised into age <60 years and age ≥60 years, ethnicity into White, Black, Asian and Minority Ethnic (BAME) groups, education into university education or no university education, and religion divided into three categories: no religion, Christianity, or other religion. Small numbers of respondents from certain ethnic groups and from religions other than Christianity or none limited subgroup analysis. Small numbers and single participant responders in some groups risked identification: we were therefore required to combine Islam, Hinduism, Judaism, Buddhism, and Sikhism as ‘religions other than Christianity’ for analysis.

### 2.4. Ethical Approval and Consent

We received NHS Research Ethics Committee (REC) (REC reference 17/LO/1602) and Health Research Authority (HRA) approval. A consent form formed the first page of the questionnaire. The study was funded by a Kidney Research UK Project Grant (RP_028_20170302). The clinical and research activities being reported are consistent with the Principles of the Declaration of Istanbul as outlined in the ‘Declaration of Istanbul on Organ Trafficking and Transplant Tourism’.

## 3. Results

A total of 1240 questionnaires were returned from 3103 patients (40% response). Participant characteristics are reported in Table 1.

LDKT recipients were more likely to respond than DDKT recipients (46% vs. 34%) and women were more likely to respond than men (43% vs. 37%) (Appendix A). However, the study participants were a population representative sample (Appendix A). Overall, the proportion of missing data was small (<3% for belief questions and <10% for all demographic variables) (Appendix A).

### 3.1. Comparison of LDKT and DDKT Recipients

DDKT recipients expressed greater uncertainty than LDKT recipients regarding all belief statements, with a greater proportion of DDKT than LDKT recipients selecting ‘Don’t know’ for every question (Table 2).

The direction of belief for DDKT and LDKT recipients was the same for nine statements (Table 2). The majority of both DDKT and LDKT recipients agreed with the statements: (1) It is morally acceptable to take a kidney from a healthy person; (3) Donating a kidney is a rewarding experience for live donors; (5) A living-donor kidney transplant may strengthen the relationship between the donor and recipient; (8) It is acceptable for a parent to receive a kidney from his/her child (over 18 years old); (9) Decisions about donation should be made by the donor alone. The recipient should not ask for a kidney. The majority of both DDKT and LDKT recipients disagreed that: (4) Donating a kidney to someone requires an extremely close personal relationship; (10) Since the donor operation is not risk free, someone who needs a kidney transplant should wait for a kidney from someone who has died. For these seven statements, DDKT and LDKT recipients who indicated that they had a belief (rather than did not know) reported the same direction of belief but for all questions a greater proportion of LDKT recipients indicated a stronger belief than DDKTs.

No difference between DDKT and LDKT recipients was found with either direction or strength of belief with respect to Statement (3)—‘Asking someone to donate makes the recipient seem selfish’. Statement (6)—‘Approaching a potential donor who then says no will change the relationship between the two people’—was associated with the greatest uncertainty for all participants; 36% of DDKT recipients and 34% of LDKT recipients selecting ‘Don’t know’ for this question.

DDKT and LDKT recipients differed in the direction of their belief with respect to only one statement. For statement (2)—‘Donors often agree to donate due to feelings of guilt or family pressure’—the majority of LDKT recipients disagreed whilst DDKT recipients were split between disagreement, agreement and not knowing (Table 2).

### 3.2. Predictors of Case–Control Status

The strength of agreement with seven belief statements predicted case–control status, even after adjustment for potential confounders (Table 3). A greater level of agreement with statements 1, 3, 5, and 8 predicted being an LDKT over a DDKT recipient. These statements concern the ‘acceptability’ of living donation and transplantation, and its ‘positive effects’ (‘rewarding experience’ and ‘strengthening relationship’). A greater level of disagreement with statements 2, 6 and 10 predicted being an LDKT over a DDKT recipient. These statements relate to beliefs about individuals experiencing ‘pressure to donate’ and the ‘risks/negative impacts of living donation’.

### 3.3. Participant Characteristics and Beliefs (Table S3a–e)

#### 3.3.1. Sex

For only one of the ten statements, responses from women and men differed. The majority of women and men agreed with Statement 8—‘It is acceptable for a parent to receive a kidney from his/her child (over 18 years old)’—but a greater proportion of women disagreed compared to men (14% versus 8%, chi^2^
*p*-value < 0.001 across all categories of agreement).

#### 3.3.2. Age

For four of the ten statements, older respondents indicated greater uncertainty by selecting ‘Don’t know’ rather than indicating a direction of belief. Individuals aged ≥60 years were more likely than individuals aged <60 years to answer ‘Don’t know’ for statement (2)—‘Donors often agree to donate due to feelings of guilt or family pressure’ (36% versus 24%, chi^2^
*p*-value < 0.001 across all categories of agreement), statement (5)—‘A living-donor kidney transplant may strengthen the relationship between the donor and recipient’ (23% versus 16%, chi^2^
*p*-value 0.02 across all categories of agreement), statement (6)—‘Approaching a potential donor who then says no will change the relationships between the two people’ (41% versus 31%, chi^2^
*p*-value < 0.001 across all categories of agreement), and statement (7)—‘Asking someone to donate makes the recipient seem selfish’ (32% versus 18%, chi^2^
*p*-value < 0.001 across all categories of agreement).

For one statement, statement (9)—‘Decisions about donation should be made by the donor alone. The recipient should not ask for a kidney’—the direction of belief differed with age. People aged ≥60 years were much more likely to agree compared to people aged <60 years (73% versus 57%, chi^2^
*p*-value < 0.001 across all categories of agreement).

#### 3.3.3. Education

For two of the ten statements, a greater proportion of those who did not go to university disagreed with the statement compared to those who did: statement (5)—‘A living-donor kidney transplant may strengthen the relationship between the donor and recipient’ (13% vs. 7%, chi^2^
*p* = 0.008), and statement (6)—‘Approaching a potential donor who then says no will change the relationship between the two people’ (49% versus 42%, chi^2^
*p*-value 0.03). For statement (9)—‘Decisions about donation should be made by the donor alone. The recipient should not ask for a kidney’—individuals without a university degree were more likely to agree than those with (66% versus 58%, chi^2^
*p*-value 0.04).

Individuals without a university degree indicated greater uncertainty with respect to statement (7)—‘Asking someone to donate makes the recipient seem selfish’—with a higher proportion selecting ‘Don’t know’ compared to those with a university degree (26% versus 18%, chi^2^
*p*-value 0.01).

#### 3.3.4. Ethnicity

The majority of both white and non-white individuals agreed with statement (1) regarding the moral acceptability of taking a living-donor transplant (89% and 81%), but of the remainder, non-white individuals were more likely to select ‘Don’t know’ than white individuals (13% versus 6%, chi^2^
*p* value = 0.002). Statement (10)—‘Since the donor operation is not risk free, someone who needs a kidney transplant should wait for a kidney from someone who has died’—generated the greatest ethnic difference in opinion: non-white individuals were less likely to say they disagreed with this statement (69% versus 85%) and more likely to indicate that they did not know (21% versus 9%, chi^2^
*p* < 0.001).

#### 3.3.5. Religion

For statement (1)—‘It is morally acceptable to take a kidney from a healthy person’—a greater proportion of people from the ‘Other religions’ group selected ‘Don’t know’ (13%) compared to those of no religion (5%) and Christians (7%) (Chi^2^
*p* = 0.01). Similarly, for statement (3)—‘Donating a kidney is a rewarding experience for the live donors’—individuals from the group comprising religions other than Christianity were less likely to agree, and more likely to select ‘Don’t know’ (24%) compared to those of no religion (19%) and Christians (11%) (Chi^2^
*p* < 0.001). For statement (10)—‘Since the donor operation is not risk free, someone who needs a kidney transplant should wait for a kidney from someone who has died’—a smaller proportion of people in the ‘Other religions’ group said that they disagreed with this statement (65%) compared to people of no religion (89%) or Christians (89%), and a greater proportion selected ‘Don’t know’ (24%) compared to Christians (10%) and people with no religion (8%) (chi^2^
*p* < 0.001).

For statement (6)—‘Approaching a potential donor who then says no will change the relationship between the two people’—a slightly greater proportion of Christians (49%) disagreed with the statement compared those of ‘Other religions’ (43%) or none (42%) (chi^2^
*p* = 0.008).

## 4. Discussion

In this questionnaire-based case–control study, we compared the beliefs of LDKT and DDKT recipients about the acceptability of living kidney donation and transplantation. We found no evidence that DDKT recipients hold strong beliefs against living-donor kidney transplantation. Rather, DDKT recipients hold similar beliefs to LDKT recipients, but report less conviction and greater uncertainty. We did not investigate the source of beliefs in this questionnaire, but it would be interesting to investigate whether the greater uncertainty in the DDKT respondents influences or reflects the beliefs of family members and potential donors. Uncertainty may reflect differing or conflicting beliefs within a family regarding the acceptability of living-donor kidney transplantation.

We aimed to identify groups with beliefs against living-donor kidney transplantation that may explain observed sex, age, ethnic and socioeconomic disparities in the uptake of LDKTs. Overall, we did not find any evidence of significant difference in the direction of belief with sex, age, ethnicity, religion or education. This suggests that inequality in LDKT uptake associated with sex, age, ethnic, or socioeconomic position is not explained by disproportionately high numbers of individuals in these groups holding beliefs that are incompatible with living-donor kidney transplantation.

BAME group ethnicity and having a religious affiliation other than Christianity were both associated with greater uncertainty regarding a number of belief statements. BAME individuals were particularly uncertain as to whether one should wait for a DDKT, given that living kidney donation is not risk free. Uncertainty regarding organ donation and transplantation has previously been reported in qualitative research amongst certain ethnic and religious groups, attributed specifically to uncertainty regarding religious edicts [31,32]. One qualitative study from the Netherlands identified a lack of awareness about the ‘official’ position of an individual’s religion regarding living organ donation within communities, and confusion due to differing interpretations of religious texts [32]. Research from the USA has shown that, amongst church-attending African-American individuals without kidney disease, 37% disagreed with living donation [33], and members of the clergy were more likely to express reservations about living donation than deceased donation (33.3% versus 16.7%) [33]. These studies suggest that faith leaders might play an important educational role, that their opinion might be influential, and that clarity over the position of the religion on living-donation needs to be made explicit [32,33,34]. To this end, during the preparation of this manuscript, a new fatwa clarifying Islamic approval of living and deceased organ donation and transplantation was published in the UK [35].

Older people reported greater uncertainty in their beliefs about the impact of donation on the family, and whether asking is selfish on the recipient’s part. Older people have been reported as being unhappy to accept an organ from a younger living donor [36,37], in part due to parents believing they should protect their children from harm [36,37]. This belief regarding the acceptability of living-donor kidney transplantation might be influenced by clinicians: research from the USA has suggested that eligible older people with kidney disease are less likely to be encouraged to seek a transplant by their nephrologists [38].

Our findings suggest that the majority of DDKT recipients believe living kidney donation and living-donor kidney transplantation are acceptable, appropriate and justifiable. The majority of demographic groups believe that there are benefits from LDKTs to both the donor and the recipient. Given these beliefs, it suggests that there is capacity to increase LDKT activity in the UK. There should be no assumption that people of certain groups (BAME or older people) have strong beliefs against an LDKT—but rather, any uncertainty should be taken as an opportunity to engage in discussion. Attitudes towards living kidney donation are often open to change and, accordingly, can be influenced [39]. Conversations with religious leaders may help to overcome specific uncertainties regarding a particular religion’s position on living donation [34,35].

The belief statements in this study were first developed and used in a Canadian population [29]. LDKT recipients and wait-listed patients surveyed in Canada were found to have the same direction of response as LDKT recipients and DDKT recipients in the UK for all statements except for Statements (4) and (10). For Statement (4)—‘Donating a kidney to someone requires an extremely close personal relationship’—69% Canadian LDKT recipients agreed or strongly agreed with this statement, compared to 26% of UK LDKT recipients. For statement (10)—‘Since the donor operation is not risk free, someone who needs a kidney transplant should wait for a kidney from someone who has died’—a greater proportion of UK DDKT recipients disagreed with this statement when compared to Canadian wait-listed patients (72% versus 52%). These differences may reflect transplant practice and beliefs changing over time, since the Canadian study was undertaken over 15 years earlier. However, these differences may in part explain why the UK’s LDKT activity is greater than Canada’s [40], and this requires further investigation.

In our study, statement (10)—‘Since the donor operation is not risk free, someone who needs a kidney transplant should wait for a kidney from someone who has died’—generated the most difference in opinion; therefore, how beliefs will change with the UK’s move to an opt-out deceased-donation law in 2020 will need to be investigated.

This was a large, multicentre study. To our knowledge, this is the first quantitative study to investigate beliefs about living-donor kidney transplantation amongst transplant recipients. The questionnaire was evaluated in cognitive interviews prior to use, validated and then piloted [26]. The proportion of missing data was small. However, the study has limitations: (i) Although our response rate was reasonable for an unincentivized postal survey, and compares to the response rate of other postal surveys in the UK [41,42] and that of previous a previous European transplant survey [43], there is a risk of self-selection bias. We have reported in our results that our population appeared population representative (Appendix A). In addition, we compared our findings to those from the Access to Transplantation and Transplant Outcome Measures (ATTOM) study (which had 72% participation), and found the same effect sizes between socioeconomic position and likelihood of an LDKT (see Appendix A) providing further evidence our sample is fairly representative of the total population of such patients. (ii) A total of 14% of participants were from BAME groups—this is not a surprising finding as in the UK between 2013 and 2017 BAME individuals comprised 17% of LDKT recipients and 27% of LDKT kidney transplant recipients [44], but this did prevent the analysis of individual ethnic groups (e.g., Asian, Black, Chinese).

The questionnaire was administered to LDKT and DDKT transplant recipients, both of whom have experienced transplantation; thus in the analyses examining the relationship between beliefs and transplant type, one might expect responses to be subject to a range of cognitive biases, including justifying their decision, and endowment effects. However, evidence against a significant endowment effect on the direction of belief includes the finding that the majority of DDKT recipients expressed positive beliefs about living donation and transplantation. Were there significant endowment effects, we would not have expected the majority of DDKT recipients to express positive beliefs about LDKTs. Cognitive biases do not explain the differences in beliefs between different demographic groups.

## 5. Conclusions

The majority of both DDKT and LDKT recipients across all demographic groups reported holding positive beliefs about living donation and transplantation. This encouraging finding suggests that, at least on the part of the transplant candidate, beliefs that are incompatible with LDKT are not a major barrier to living-donor transplantation in the UK, and that there is capacity to increase LDKT activity.

## Figures and Tables

**Table 1 jcm-09-00031-t001:** Participant characteristics by case–control status ^a^.

Characteristics		Cases ^b^ (LDKTs) *n* = 672	Controls ^b^ (DDKTs) *n* = 565	Chi^2^ Comparing Cases and Controls
**Sex**	Male	382 (57)	322 (57)	*p* = 0.95
	Female	279 (42)	235 (42)	
	Missing	11 (2)	8 (1)	
**Age (years)**	20–29	47 (7)	27 (5)	*p* = 0.39
	30–39	80 (12)	57 (10)	
	40–49	106 (16)	102 (18)	
	50–59	178 (27)	153 (27)	
	60–69	167 (25)	132 (23)	
	>70	77 (12)	79 (14)	
	Missing	17 (3)	15 (3)	
**Ethnicity**	White	581 (87)	445 (79)	*p* = 0.005
	Asian	38 (6)	41 (7)	
	Black/African/Caribbean	19 (3)	39 (7)	
	Mixed/Multiple	5 (0.7)	5 (0.9)	
	Other	10 (2)	14 (3)	
	Missing	19 (3)	21 (4)	
**Religion**	No religion	191 (28)	144 (26)	*p* = 0.01
	Christian	402 (60)	315 (56)	
	Muslim	10 (2)	11 (2)	
	Other religions ^c^	37 (6)	56 (10)	
	Missing	22 (3)	39 (7)	
**Highest level of education**	No formal education/Primary school	10 (2)	20 (4)	*p* = 0.03
	Secondary school	202 (30)	191 (34)	
	Vocational/Technical	171 (26)	143 (25)	
	University-undergraduate	145 (22)	98 (17)	
	University-postgraduate	73 (11)	46 (8)	
	Other	33 (5)	24 (4)	
	Missing	38 (6)	43 (8)	

^a^ The three participants for whom transplant type/case–control status was missing are excluded from this table. ^b^ Percentages may not total 100% due to figures being presented to the nearest whole number. ^c^ Hindu, Jewish, Sikh, Buddhist and Other combined due to single participant responders in some groups risking identification.

**Table 2 jcm-09-00031-t002:** Beliefs about living donation and living-donor kidney transplantation.

Belief Statement	Transplant Type	Strongly Disagree *n* (%)	Disagreen *n* (%)	Agreen *n* (%)	Strongly Agree *n* (%)	Don’t Know *n* (%)	Chi^2^ *p*-Value
1. It is morally acceptable to take a kidney from a healthy person.	DDKT ^a^	8 (2)	22 (4)	293 (53)	172 (31)	52 (10)	<0.001
	LDKT ^a^	24 (4)	11 (2)	252 (39)	340 (52)	28 (4)	
2. Donors often agree to donate due to feelings of guilt or family pressure.	DDKT	63 (11)	172 (31)	117 (21)	22 (4)	177 (32)	<0.001
	LDKT	134 (20)	262 (40)	81 (12)	9 (1)	170 (26)	
3. Donating a kidney is a rewarding experience for the live donors.	DDKT	6 (1)	5 (0.9)	260 (47)	158 (29)	123 (22)	<0.001
	LDKT	11 (2)	4 (0.6)	269 (41)	314 (48)	60 (9)	
4. Donating a kidney to someone requires an extremely close personal relationship.	DDKT	78 (14)	286 (52)	79 (14)	47 (9)	62 (11)	0.004
	LDKT	121 (18)	331 (50)	110 (17)	59 (9)	38 (6)	
5. A living-donor kidney transplant may strengthen the relationship between the donor and recipient.	DDKT	8 (2)	55 (10)	254 (46)	77 (14)	158 (29)	<0.001
	LDKT	13 (2)	65 (10)	314 (48)	198 (30)	69 (11)	
6. Approaching a potential donor who then says no will change the relationship between the two people.	DDKT	47 (9)	185 (34)	85 (16)	33 (6)	200 (36)	0.001
	LDKT	91 (14)	235 (36)	96 (15)	14 (2)	222 (34)	
7. Asking someone to donate makes the recipient seem selfish.	DDKT	45 (8)	204 (37)	120 (22)	41 (8)	139 (25)	0.45
	LDKT	68 (10)	256 (39)	145 (22)	38 (6)	151 (23)	
8. It is acceptable for a parent to receive a kidney from his/her child (over 18 years old).	DDKT	23 (4)	45 (8)	292 (53)	106 (19)	86 (16)	0.002
	LDKT	17 (3)	40 (6)	365 (56)	169 (26)	68 (10)	
9. Decisions about donation should be made by the donor alone. The recipient should not ask for a kidney.	DDKT	19 (4)	112 (20)	203 (37)	127 (23)	90 (16)	<0.001
	LDKT	42 (6)	121 (18)	213 (32)	220 (33)	62 (9)	
10. Since the donor operation is not risk free, someone who needs a kidney transplant should wait for a kidney from someone who has died.	DDKT	87 (16)	311 (56)	52 (9)	10 (2)	92 (17)	<0.001
	LDKT	265 (40)	336 (51)	5 (0.8)	8 (1)	44 (7)	

^a^ DDKT = deceased-donor kidney transplant; LDKT = living-donor kidney transplant.

**Table 3 jcm-09-00031-t003:** Strength of agreement and likelihood of being an LDKT recipient over a DDKT recipient.

Belief Statement	Association between Agreement with Statement and Likelihood of Being an LDKT Recipient over a DDKT Recipient		Interpretation
	Unadjusted OR (95% CI)	Adjusted OR ^a^ (95% CI)	
1. It is morally acceptable to take a kidney from a healthy person.	1.47 (1.26–1.71)	1.47 (1.29–1.68)	Agreement with statement predicts being an LDKT recipient
2. Donors often agree to donate due to feelings of guilt or family pressure.	0.56 (0.45–0.70)	0.57 (0.45–0.73)	Disagreement with statement predicts being an LDKT recipient
3. Donating a kidney is a rewarding experience for the live donors.	1.56 (1.24–1.94)	1.42 (1.13–1.78)	Agreement with statement predicts being an LDKT recipient
4. Donating a kidney to someone requires an extremely close personal relationship.	0.97 (0.84–1.13)	0.94 (0.79–1.12)	
5. A living-donor kidney transplant may strengthen the relationship between the donor and recipient.	1.42 (1.20–1.68)	1.45 (1.21–1.74)	Agreement with statement predicts being an LDKT recipient
6. Approaching a potential donor who then says no will change the relationship between the two people.	0.69 (0.62–0.78)	0.62 (0.55–0.71)	Disagreement with statement predicts being an LDKT recipient
7. Asking someone to donate makes the recipient seem selfish.	0.88 (0.75–1.02)	0.86 (0.71–1.04)	
8. It is acceptable for a parent to receive a kidney from his/her child (over 18 years old).	1.31 (1.10–1.56)	1.29 (1.04–1.60)	Agreement with statement predicts being an LDKT recipient
9. Decisions about donation should be made by the donor alone. The recipient should not ask for a kidney.	1.09 (0.98–1.21)	1.05 (0.95–1.19)	
10. Since the donor operation is not risk free, someone who needs a kidney transplant should wait for a kidney from someone who has died.	0.36 (0.27–0.47)	0.38 (0.27–0.54)	Disagreement with statement predicts being an LDKT recipient

^a^ Adjusted for sex, 10-year age-group, ethnicity (White and Black, Asian and Minority Ethnic (BAME) groups), religion (No religion, Christian, Other), university education (university education or no university education.

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
