# Peer review of "Beliefs of UK Transplant Recipients about Living Kidney Donation and Transplantation: Findings from a Multicentre Questionnaire-Based Case–Control Study"

_jcm, 2019, doi:10.3390/jcm9010031_

Round 1
Reviewer 1 Report
The authors are presenting a very interesting survey. It is an important issue and difficult to address, yet they managed to do it with a thoughtful study design.
In tables, it is unclear what is compared and "p" is refering to.
The experimental section is a bit confusing and crowded, i would recommend dividing it into sub-sections.
adding a box with questionnaries regarding participant demographics and characteristics would be helpful.
In results, uncertainty seems to be much higher in DDKT, yet this point in not addressed in conclusion.
It is surprising that the number of patients in subgroups such as BAME are small in the UK. It would be interesting to extend this survey on to patients on waiting list and compare those with recipients.
I hope the authors are planing to follow up on the new fatwa clarification and its impact on LDKT.
This article shows valuable results, confirming that improving the enlightenment for LDKT is needed, regardless of ethnicity or Religion.
Reviewer 2 Report
Well-written article on the perception of both deceased and live donor renal transplant recipients. This article explains some of the beliefs and misconceptions that many have regarding the less utilization of living donor transplantation throughout the world.Well used statistical methods.
Reviewer 3 Report
The authors present findings of a questionnaire taken among kidney transplant recipients. The study answers an interesting research question and appears to be well designed. The data are relevant for the transplant community, but the clinical impact may be somewhat limited. The questionnaire the authors used seems apt to answer the study questions and is based on previous literature. The study population is representative of the British transplant population. Also, the authors show an extensive missing-data analysis and explain very well how they minimized the risk of self-selection bias. Overall, this is a great study. I have some suggestions to further improve the quality of the manuscript:
Experimental section, page 2, line 56 and on. The authors state “which is why transplant recipients were selected as participants rather than people with Chronic Kidney Disease or those on dialysis, some of whom may not believe transplantation is acceptable at all.”. This assumption may cause some confusion among readers. Consider rephrasing this sentence. The authors state “PB performed stratified random sampling using Stata 15 [25] to select on average 110 LDKTs and 110 DDKTs from each site, weighted by the number of transplants performed annually at each study site”. Do the authors mean “Pippa Bailey”? Consider removing the authors' name from this section. The authors state: “The calculation indicated that 170 patients would be needed and that 944 would be needed to allow analyses stratified by Index of Multiple Deprivation rank quintile and allow for 10% missing data. This sample size allows detection of a far smaller difference (0.16 Standard Deviation) for a dichotomous exposure or between 6-8% for a categorical outcome [26].” Do the authors mean: 944 patients in each group (LDKT/DDKT) or in total? The relevance of the Chi2 test on case-control characteristics may not be relevant without additional analysis: which characteristics specifically driver differences between LKDT and DDKT? Since groups are small, the difference in the Chi2 test may not be relevant. Table 2: it may be more informative to report beliefs as “number (%)”, instead of % alone. I suggest that the authors edit the first paragraph of their discussion, to make it more in line with the introduction. In their discussion, the authors state: “It would be interesting to investigate whether and how the weaker conviction of certain individuals regarding the acceptability and positive effects of living-donor transplantation influences the decision-making of potential donors to donate a kidney, as well as the extent to which a recipient’s beliefs are influenced by family members.”. This part seems out of place. After this sentence, they start mentioning their data again. It may be beneficial to turn this around to make a leaner discussion section. Discussion, page 2, line 70+71, the authors state: “…it was not possible to effectively dissociate ethnic origin from religious orientation in this study.”. It is not completely clear what the authors mean by this from the sentence, consider making it more concrete. It may also be worthwhile to compare British numbers vs USA numbers. In their discussion, the authors compared LDKT/DDKT response in the UK to an older (2005) study in the Canadian population; the population that the belief statements were developed in. Can changes in Transplant practice over time also explain the results? Unfortunately, living kidney donors themselves were not included in this study. If there is any possibility to include them, it would greatly enhance the quality of the results. In their discussion the authors state: “… that beliefs that are incompatible with LDKT are not a major barrier to living-donor transplantation in the UK, and that there is capacity to increase LDKT activity.”. Without including living donors in the analysis, this statement might be a little too strong.
Author Response
Please see the attachment.

This manuscript is a resubmission of an earlier submission. The following is a list of the peer review reports and author responses from that submission.